# Legal Judgment Prediction Based on Machine Learning: Predicting the Discretionary Damages of Mental Suffering in Fatal Car Accident Cases

**Decheng Hsieh [1,*], Lieuhen Chen [1] and Taiping Sun [2]**

[1]  Department of Computer Science and Information Engineering, College of Science and Technology, National Chi Nan University, Nantou County 545301, Taiwan; lhchen@csie.ncnu.edu.tw

[2]  Department of Electrical Engineering, College of Science and Technology, National Chi Nan University, Nantou County 545301, Taiwan; tps@ncnu.edu.tw

*  Correspondence: a97141481@gmail.com

**Abstract:** The discretionary damage of mental suffering in fatal car accident cases in Taiwan is subjective, uncertain, and unpredictable; thus, plaintiffs, defendants, and their lawyers find it difficult to judge whether spending much of their money and time on the lawsuit is worthwhile and which legal factors judges will consider important and dominant when they are assessing the mental suffering damages. To address these problems, we propose k-nearest neighbor, classification and regression trees, and random forests as learning algorithms for regression to build optimal predictive models. In addition, we reveal the importance ranking of legal factors by permutation feature importance. The experimental results show that the random forest model outperformed the other models and achieved good performance, and "the mental suffering damages that plaintiff claims" and "the age of the victim" play important roles in assessments of mental suffering damages in fatal car accident cases in Taiwan. Therefore, litigants and their lawyers can predict the discretionary damages of mental suffering in advance and wisely decide whether they should litigate or not, and then they can focus on the crucial legal factors and develop the best litigation strategy.

**Keywords:** discretionary damages of mental suffering; fatal car accident cases; legal judgment prediction; mental suffering damages; relevant legal factors

## 1. Introduction

In cases where a victim is killed in a car accident, the father, mother, sons, daughters, and spouse of the deceased may claim for reasonable mental suffering damages in accordance with Article 194 of the Taiwan Civil Code. However, the mental suffering damages that come with losing someone are often difficult to calculate, and no standard formula exists [1]; therefore, judges need to consider numerous legal factors which have been indicated by the Taiwan Supreme Court to assess a specific dollar amount on the mental suffering damages [2]. In other words, the assessment of reasonable mental suffering damages is very subjective and unpredictable, and it will cause many serious problems. To begin with, two parties might find it difficult to predict the discretionary damages of mental suffering made by judges; thus, they will wonder whether spending much of their money and time on the lawsuit is worthwhile. Even though they decide to engage in a lawsuit, it is impossible for them to prepare evidentiary documents and materials about relevant legal factors, because they do not know which legal factors judges will consider important and dominant when they are assessing the mental suffering damages, so they might focus their efforts on irrelevant legal factors rather than relevant legal factors. Even professional lawyers who are familiar with past cases may be frustrated by those problems because they are suffering from the shortcomings of human reasoners [3]. Specifically, these serious problems may prevent two parties from dealing with their controversies by

the due process of law and make them choose illegal ways to do so. Furthermore, those problems might make plaintiffs abandon their rights in civil affairs and compensation for damages. In summary, it is important and necessary to use machine learning techniques to predict the discretionary damages of mental suffering in fatal car accident cases in Taiwan and reveal the importance ranking of legal factors so that those problems and difficulties can be solved once and for all.

Recently, there have been many studies on legal judgment predictions based on machine learning (ML) techniques. Some studies have used structuring textual data, which often implement natural language processing (NLP) techniques. For example, Li et al. used the conditional random field (CRF) method to fetch legal factor labels of robbery and intimidation cases and proposed an additive regression model to predict the sentencing in such cases [4]. In addition, Aletras et al. used N-gram to obtain features from the European Court of Human Rights judgments and proposed a support vector machine (SVM) with linear kernel to predict whether a case violated an article of the Convention on Human Rights [5]. A similar study obtained N-gram features from the case text for the Supreme Court of the United States and the United States Circuit Court and built ML and deep learning (DL) models to predict those appellate affirm or reverse decisions, and a convolutional neural network (CNN) model for district-to-circuit reversal prediction outperformed other models [6]. In addition, Li et al. used NLP techniques to build a knowledge extraction engine and obtain a database, proposing a Markov logic network to predict the judicial decision of divorce cases [7]. Furthermore, Jiang et al. used a deep reinforcement learning method to extract rationales from input text and implemented a charge prediction task [8]. Moreover, Chen et al. established a legal graph network to fuse complete charge information into a unified legal graph and used an attention-based neural network for charge prediction [9]. In addition, Zhang et al. used bidirectional encoder representations from transformers (BERTs) to label the sentences of legal factors in the judgments and proposed CNN, long short-term memory (LSTM), and a gated recurrent unit (GRU) to extract features and predict the sentencing class [10].

In contrast, several studies have used features available directly from established datasets rather than textual documents to implement legal judgment prediction; thus, they did not use NLP techniques to extract features. For example, Katz et al. used a dataset from the Supreme Court database and proposed random forests (RFs) to predict the voting behavior of the Court and its Justices [11]. In addition, Huang et al. focused on child custody cases after divorce in Taiwan and used a tabular dataset made by legal experts to propose artificial neural networks (ANNs), decision trees, and gradient boosting to predict whether the father or the mother should receive custody. Moreover, they also demonstrated some specific features that judges considered important [12–14]. In addition, Franca et al. used a private database provided by an energy supply company and proposed tree-based gradient boosting to predict the outcomes of energy market lawsuits [15].

Despite these efforts in legal judgment prediction based on ML techniques, most previous studies mainly focused on classification tasks instead of regression tasks, and there has been no research using ML techniques to predict the discretionary damages of mental suffering in fatal car accident cases in Taiwan and unveil the importance ranking of legal factors by permutation feature importance. In order to solve the above challenges, in this study, we propose three basic but well-known learning algorithms to predict the discretionary damages of mental suffering in fatal car accident cases in Taiwan, including k-nearest neighbor (KNN), classification and regression trees (CART), and RF, to determine whether basic and popular ML algorithms are good enough to address our prediction task and define future work according to the final results. In addition, we used permutation feature importance to reveal the importance ranking of legal factors. In addition, there was no need to implement NLP techniques, because we used a dataset made by legal experts, which contained features extracted by them.

However, we faced two major difficulties in our work. To start with, many judges refuse to unveil the true values of legal factors due to the protection of rights to privacy,

which strictly limited the amount of data in our dataset. Moreover, the range of discretionary damages of mental suffering in our dataset was very wide, i.e., from 15 to 500 in TWD ten thousand, which might have affected the model learning and decrease model performance. In addition, we hypothesized that the RF model should outperform other models, because the RF belonged to ensemble learning, which is often considered to be more robust than a single base learner [16]. Furthermore, we infer that "the mental suffering damages that plaintiff claims" should have higher importance because it represents the disposition principle that all Taiwanese judges must abide by when they are assessing mental suffering damages.

To summarize, the experimental results show that the RF model outperformed other models and achieved good performance; therefore, litigants and their lawyers could predict the discretionary damages of mental suffering in advance, and they can easily break the uncertainty of judicial outcomes and wisely decide whether they should litigate or not. Furthermore, we have successfully revealed the importance ranking of legal factors, so legal actors can focus on the crucial factors and develop the best litigation strategies. For example, they can concentrate on finding evidentiary documents and materials about the most important or dominant legal factors instead of wasting their precious time and money on irrelevant legal factors. On the other hand, this methodology can be applied to other types of mental suffering damages prediction tasks in Taiwan with minor and suitable adjustments, including mental suffering from physical injuries or pain, humiliation, privacy infringements, and divorces. Moreover, people can claim mental suffering damages and judges need to consider several factors such as the circumstances of illegal infringement and earning capability of two parties, in accordance with the Interpretation of the Supreme People's Court on Problems regarding the Ascertainment of Compensation Liability for Emotional Damages in Civil Torts [17], which is similar to the assessment of mental suffering damages in Taiwan; thus, our methodology could be applied to the region of mainland China after suitable adjustments.

This study proceeds as follows. In Section 2, we discuss the dataset and the approaches we used in this research. Section 3 presents the experimental results and discussion. Finally, in Section 4, we provide the conclusions and future work of this research.

## 2. Materials and Methods

The overall goal of this study was to build the optimal model to predict the discretionary damages of mental suffering in fatal car accident cases in Taiwan and reveal the importance ranking of legal factors. In order to achieve these goals, our approach involved the following steps. First, we implemented data preprocessing, including feature encoding, imputing missing values, feature scaling, and feature selection. Next, we used three basic and classic ML algorithms for regression to train predictive models, and then implemented hyperparameter tuning. Finally, we calculated the importance ranking of legal factors based on those three models by permutation feature importance.

In this study, we used the scikit-learn package (version 0.24.2) in Python language [18] to implement machine learning tasks and all steps involved.

### 2.1. Dataset

The Taiwan Judicial Yuan Law and Regulations Retrieving System was used to search and collect judgments from 2006 to 2020 in Taiwan Taichung District Court, and irrelevant judgments, such as judgments against the plaintiff (without assessing the mental suffering damages), were removed. Furthermore, this study only focused on judgments with one victim and one defendant, excluding judgments with more than one victim or defendant.

Next, we took one plaintiff as one data unit [2], extracted the legal factors from the judgments, and used them as features in the dataset, as seen on Figure 1. Apart from the legal factors which judges considered while assessing mental suffering damages, we added some extra features which we deemed important and relevant to the assessment of mental suffering damages. For example, we added some features that might affect the

discretionary damages, including "the total number of plaintiffs who are victim's parents in the same case", "the total number of plaintiffs who are victim's children in the same case", and "the total number of plaintiffs in the same case".

**Figure 1.** An example of how we extracted legal factors from the judgment documents.

Subsequently, we extracted discretionary damages from the judgments and treated them as the target values ($y$ vector) in the dataset. Finally, there were some features in the dataset lacking the unified unit; therefore, we determined the unified unit for those feature columns. Above all, we eliminated the name of plaintiffs, defendants, and judges to protect their privacy, and we also covered the number of judgments for fear that someone might find out their personal information and violate their right to information privacy by surfing the Taiwan Judicial Yuan Law and Regulations Retrieving System website. In summary, the dataset contained 34 features, which are shown in Table 1.

In the total 483 observations in the dataset, the range of the predicted values was from 15 to 500 in TWD ten thousand, and we created a 70/30 train–test split of the dataset, i.e., we randomly split the dataset into a training set with 338 samples and a test set with 145 samples. Furthermore, we set the random_state parameter of the train_test_split to 0 in order to randomly shuffle the data before splitting.

The whole dataset we used in this research is available from the following link: https://zenodo.org/record/5565766#.YW0X19lBwnV (accessed on 3 November 2021).

### 2.2. Data Preprocessing

In the data preprocessing part, we implemented feature encoding and imputed missing values, and then used standardization and feature selection to achieve feature scaling and dimensionality reduction.

#### 2.2.1. Feature Encoding, Imputing Missing Values, and Feature Scaling

The categorical features in the dataset were not numeric; therefore, we needed to turn them into numbers [19]. For the ordinal categorical features such as "the plaintiff's educational background" and "the defendant's educational background", ordinal coding was used, which meant that each category was assigned an integer number [20] based on the prior information [21]. For the nominal categorical features such as "the victim is in poor health", "the plaintiff is in debt", and "the defendant is in debt", one-hot encoding [19] was used. In theory, one-hot encoding takes each of the categories as a feature: a categorical feature with k categories is encoded as a feature vector of length k [19]. However, all nominal categorical features in our dataset shown in Table 1 were encoded as a feature vector of length 1, because those features were all binary categorical features.

**Table 1.** The description of features in dataset.

| Serial Number | Feature | Type |
|---|---|---|
| 1 | The total number of plaintiffs who are the victim's parents in the same case | Numerical |
| 2 | The total number of plaintiffs who are the victim's spouse in the same case | Numerical |
| 3 | The total number of plaintiffs who are the victim's children in the same case | Numerical |
| 4 | The total number of plaintiffs in the same case | Numerical |
| 5 | The age of the victim | Numerical |
| 6 | The victim is the only child of the plaintiff | Nominal categorical |
| 7 | The victim is in poor health | Nominal categorical |
| 8 | The plaintiff is the victim's parents | Nominal categorical |
| 9 | The plaintiff is the victim's spouse (including the duration of marriage) | Ordinal categorical |
| 10 | The plaintiff is the victim's child (including the plaintiff is the only child of the victim) | Ordinal categorical |
| 11 | The number of children that the plaintiff raises | Numerical |
| 12 | The number of relatives that the plaintiff needs to support | Numerical |
| 13 | The plaintiff's educational background | Ordinal categorical |
| 14 | The plaintiff's occupation | Nominal categorical |
| 15 | The plaintiff's income | Numerical |
| 16 | The amount of real estate that the plaintiff possesses | Numerical |
| 17 | The number of automobiles that the plaintiff possesses | Numerical |
| 18 | The number of motorcycles that the plaintiff possesses | Numerical |
| 19 | The number of investments that the plaintiff holds | Numerical |
| 20 | The plaintiff is in debt | Nominal categorical |
| 21 | The plaintiff belongs to disadvantaged and vulnerable groups | Nominal categorical |
| 22 | The number of children that the defendant raises | Numerical |
| 23 | The number of relatives that the defendant needs to support | Numerical |
| 24 | The defendant's educational background | Ordinal categorical |
| 25 | The defendant's occupation | Nominal categorical |
| 26 | The defendant's income | Numerical |
| 27 | The amount of real estate that the defendant possesses | Numerical |
| 28 | The number of automobiles that the defendant possesses | Numerical |
| 29 | The number of motorcycles that the defendant possesses | Numerical |
| 30 | The number of investments that the defendant holds | Numerical |
| 31 | The defendant is in debt | Nominal categorical |
| 32 | The defendant belongs to disadvantaged and vulnerable groups | Nominal categorical |
| 33 | The sum of mental suffering damages that all plaintiffs claim in the same case (in TWD ten thousand) | Numerical |
| 34 | The mental suffering damages that plaintiff claims (in TWD ten thousand) | Numerical |

Next, it was required to handle the missing values problem with imputation methods, because most learning algorithms are, in general, unable to deal with missing values [22]. We imputed the missing values of the categorical features with the most common feature value [23] and replaced the missing values of the numerical features with the mean of the feature [24]. However, we imputed features related to two parties' personal property such as "the amount of real estate that plaintiff possesses", "the number of automobiles that plaintiff possesses", and "the number of motorcycles that plaintiff possesses", with 0 based on the inferences that judges often omit the description of personal property if they know that the number of personal properties is 0 [25].

In addition, we used standardization to implement feature scaling on the dataset for KNN, because it is very sensitive to the scale of features. Specifically, standardization can assign features in the dataset a mean of 0 and variance of 1 [26], and it is defined as [16]:

$$x_{std}^{(i)} = \frac{x^{(i)} - \mu_x}{\sigma_x} \tag{1}$$

where $x_{std}^{(i)}$ is the standard score of the $i$th sample in feature $x$, $x^{(i)}$ is the original value of the $i$th sample in feature $x$, $\mu_x$ is the mean of the feature $x$, and $\sigma_x$ is the standard deviation of the feature $x$ [16]. However, we did not implement standardization on the datasets which were used to train the tree-based models, because tools such as CART and RF are not sensitive to the scale of the features [19].

### 2.2.2. Feature Selection

We used feature selection to reduce the dimensionality of the original dataset and select the most appropriate feature subset with relevant features [27]. Specifically, we implemented feature selection with wrapper feature selection algorithms, because they can evaluate the candidate feature subsets on the learning algorithm and keep the feature subset that performs best [27]. In other words, the learning algorithm itself is used as part of the evaluation function [28].

To begin with, we used sequential forward selection (SFS) and sequential backward selection (SBS) as selection algorithms to implement feature selection for KNN and CART. SFS starts with an empty set, and it adds one feature which achieves the best performance to the empty set and forms the current subset. Then, it adds another feature which achieves the best performance from the remaining features to the current subset and forms a new current feature subset. This procedure is repeated until the stopping criterion is reached [29]. On the other hand, the SBS starts with the complete feature subset with all features, and it eliminates one feature whose elimination improves the performance the most. This process is repeated until the stopping criterion is reached [29].

Furthermore, we used 10-fold cross-validation and R-squared ($R^2$) as validation methods, and metrics of the evaluation function to evaluate the candidate feature subsets. However, we did not set any stopping criterion; as a result, the best feature subset with the best cross-validation score could be selected. After the best feature subsets were chosen through SFS and SBS, the two feature subsets were evaluated on the independent test set [28], and the best one was selected as the final feature subset.

Moreover, we used recursive feature elimination with cross-validation (RFECV) to implement feature selection for RF instead of using SFS and SBS, as the combination of sequential feature selection and RF could increase the computational complexity and cost. RFECV trains the model on the initial set of features and computes the importance of each feature. Then, it finds the feature with the smallest ranking criterion and eliminates it from the feature set so as to create a new feature subset [30]. After creating a new feature subset, RFECV will use cross-validation to evaluate the model which is trained on the new feature subset. If cross-validation shows that the performance of the model improves after eliminating the feature with the smallest ranking criterion, RFECV will continue to the next loop. This procedure is repeated until cross-validation shows that the performance of

the model becomes worse after eliminating the feature with smallest ranking criterion [21]. Furthermore, the measure of ranking criterion we used was the impurity importance based on the mean decrease in impurity which was computed on the RF model [31,32], and we used 10-fold cross-validation and $R^2$ as a validation method and metric for RFECV. However, we did not set any stopping criterion; as a result, the best feature subset with the best cross-validation score could be selected.

Finally, we mainly used the default values of the learning parameters in the scikit-learn package while implementing feature selection. However, random_state parameters of the DecisionTreeRegressor and RandomForestRegressor were set to popular integer 42 in order to produce the same results across different calls [31], and the max_features parameters of the RandomForestRegressor were set to the square root of the number of features according to the typical value of that hyperparameter [26].

### 2.3. Regression Models

This study was concentrated on the regression task; therefore, we employed three basic and classic ML algorithms for regression to build predictive models and examine whether those basic and popular ML algorithms are good enough to address our prediction task.

#### 2.3.1. K-Nearest Neighbor

The KNN algorithm is used for memory-based and instance-based learning; therefore, there is no model needing to be fitted [33]. To begin with, we need to choose the distance metric in order to measure all distances between the test sample and the training samples in the feature space. The common distance metric measures are the Euclidean distance and Manhattan distance, and they are defined as [34]:

$$\text{Euclidean distance} = \sqrt{(x - p)^2} \tag{2}$$

$$\text{Manhattan distance} = |x - p| \tag{3}$$

where $x$ is a training sample and $p$ is the test sample [34].

Next, we needed to select the value of k to find the k closest training samples (neighbors) that were closest in distance to the test sample; then, we could compute the predicted value for regression by returning the average value of the k neighbors [35]. However, the predicted value mentioned above does not take into account the distance weighting; as a result, the k neighbors have equal influence. On the other hand, we could choose distance weighting, allowing the neighbor which is closer to have more importance or influence [34]. In a word, the KNN algorithm can achieve good results in practice, and it is very simple [33].

#### 2.3.2. Classification and Regression Trees

CART is a tree-based algorithm, and it can be used for both classification and regression [36]. In other words, CARTs contain classification trees and regression trees [35]. However, we only focused on regression trees because this study aimed to predict continuous values.

To begin with, a CART is typically constrained to construct binary trees in the consideration of computational cost [35]. Specifically, CARTs will grow by means of binary recursive partitioning, i.e., the parent node will always be split into two child nodes, and then each child node will become a parent node and will be split into two child nodes. This process will continue until the node is a terminal node [37].

Moreover, CARTs will search all possible features and all possible values (split-point) to find the best split which can achieve the maximum homogeneity of the two child nodes.

In other words, choosing the best split is equivalent to the maximization of the change in impurity function, which is defined as [38]:

$$\Delta i(t) = i(t_p) - P_l i(t_l) - P_r i(t_r)$$
$$\underset{x_j \leq x_j^R, \; j=1,\,\ldots,\,M}{\arg\max} \left[ i(t_p) - P_l i(t_l) - P_r i(t_r) \right] \tag{4}$$

where $\Delta i(t)$ is the maximization of the change in impurity function, $i()$ is the impurity function, $t_p$, $t_l$, and $t_r$ are the parent node, left node, and right node, respectively, $P_l$ and $P_r$ are probabilities of the left node and right node, $x_j$ is feature $j$, $x_j^R$ is the best split value of feature $j$, and $M$ is the number of features [38]. After the best split is chosen, the samples whose value are greater than the split-point will be assigned to the right node, whereas the rest will be assigned to the left node [26].

In addition, the common impurity functions (splitting criteria) for regression trees are the mean squared error criterion and mean absolute error criterion, which are defined as [31]:

$$\overline{y}_m = \frac{1}{N_m} \sum_{y \in Q_m} y$$
$$mean\ squared\ error\ criterion = H(Q_m) = \frac{1}{N_m} \sum_{y \in Q_m} (y - \overline{y}_m)^2 \tag{5}$$

$$mean\ absolute\ error\ criterion\ H(Q_m) = \frac{1}{N_m} \sum_{y \in Q_m} |y - median(y)_m| \tag{6}$$

where $Q_m$ is the set of samples in node $m$, $N_m$ is the number of samples in node $m$, $H()$ is the impurity function, $y$ is the target values in vector $y$, $\overline{y}_m$ is the mean of $y$, and $median(y)_m$ is the median of $y$ [31].

If we take the mean squared error criterion, the predicted value can be computed as the average of all samples in the terminal node. However, the predicted value is computed as the median of all samples in the terminal node when we take the mean absolute error criterion [31]. In summary, the key advantage of the CART algorithm is its robustness to outliers and being able to isolate outliers in an individual node or nodes [38]. Moreover, its great interpretability can fully describe the feature space partition [26].

### 2.3.3. Random Forests

RFs involve ensemble learning, i.e., the RF is the ensemble of CART and can be used for classification and regression problems [39]. This research aimed to predict continuous values; therefore, we only paid attention to the regression forests. Specifically, the RF combines many trees, and each tree is built with a bootstrap sample which is randomly chosen from the training samples by sampling with a replacement. At each node of the trees, a subset of features is randomly chosen from the original feature set by sampling without replacement, and the best split is found only within the random subset of features [40]. In addition, the size of the random subset of features should be less than or equal to that of the original feature set [26].

Moreover, all trees in the forest are unpruned, i.e., they are so-called maximal trees or fully grown trees [41]. After growing a given number of trees in the regression forests, we can compute the predicted value by determining the mean predicted values of all trees in the regression forests [42].

In short, RFs are very popular and robust in practice due to the randomization of bootstrap samples and the random subsets of features [41].

### 2.4. Hyperparameter Tuning

It is very important to tune the hyperparameters of learning algorithms because the optimal hyperparameters can help to build the optimal model [43]; thus, we used grid searches to implement hyperparameter tuning and introduced the hyperparameter spaces of KNN, CART, and RF.

### 2.4.1. Grid Search

A grid search is an exhaustive search, and it can choose the best hyperparameter combination from a defined hyperparameter space [43]. Furthermore, we used 10-fold cross-validation and $R^2$ as an evaluation method and metric to evaluate the performance of all hyperparameter combinations and selected the one with the best cross-validation score [43]. In addition, we limited the search space by specifying the range of each hyperparameter with minimal value, maximal value, and number of steps [43].

### 2.4.2. The Hyperparameter Space

To begin with, we tuned the hyperparameters of KNN, including the distance metric, the value of k, and the distance weighting. The reasons why we tuned these hyperparameters are as follows. First, different distance metrics mean different ways to calculate the distance between the training samples and test point. Second, the KNN will be sensitive to noise if the value of k is too small. However, on the other hand, the accuracy will reduce if the value of k is too large, because it considers datapoints that are too far away [35]. Finally, distance weighting is used to determine in which the datapoints are treated [44].

To sum up, Table 2 shows the hyperparameter space of KNN in detail.

**Table 2.** The hyperparameter space of KNN.

| Hyperparameter | Hyperparameter Candidates |
|:---:|:---:|
| Distance metric | Euclidean distance |
| | Manhattan distance |
| The value of k | minimal value: 1 |
| | maximal value: 338 (the number of training samples) |
| | numbers of steps: 1 |
| Distance weighting | uniform weights (each neighbor is weighted equally) |
| | weighting by the inverse of neighbors' distance |

Next, we tuned the hyperparameters of CART, including the splitting criterion, the size of the tree, and the minimum node size. The reasons why we tuned these hyperparameters are as follows. First, using different splitting criterions such as mean squared error criterion or mean absolute error criterion might build different CART models, and those criterions will set different predicted values of the terminal nodes [31]. In addition, the size of the tree can affect the CART model's complexity [26], and it can prevent the tree from becoming a fully grown tree or maximal tree and stop the tree from overfitting [37]. Furthermore, the minimum node size refers to the minimum number of samples required to split an internal node, i.e., the tree will grow until a given minimum node size is reached; thus, the CART model can be pruned to an optimal size with this hyperparameter [37]. In practice, the minimum node size is often set to 10% of the number of training samples [38].

In short, Table 3 shows the hyperparameter space of CART in detail.

Furthermore, we tuned the hyperparameters of RF, including the number of trees in the forest, the size of the random subset of features, the splitting criterion, the size of the tree, and the minimum node size. The reasons why we tuned these hyperparameters are as follows.

**Table 3.** The hyperparameter space of CART.

| Hyperparameter | Hyperparameter Candidates |
|---|---|
| Splitting criterion | mean squared error criterion |
| | mean absolute error criterion |
| The size of the tree | minimal value: 2 |
| | maximal value: 17 (the size of maximal tree) |
| | numbers of steps: 1 |
| Minimum node size | minimal value: 2 |
| | maximal value: 33 (10% of the number of training samples) |
| | numbers of steps: 1 |

In general, the larger the number of trees, the better the performance will be. However, the performance will stop improving when the number of trees increases beyond a critical number [31]. Although the size of the bootstrap sample is often set to the number of original training samples [16], it is important to tune the size of the random subset of features, because they can control the degree of the randomization and reduce the variance [41]. In practice, the size of the random subset of features is usually set to the square root of the number of features [26,35] or the number of features divided by three [39]. In order to achieve the best performance, the hyperparameters of CART should be tuned, including the splitting criterion, the size of the tree, and the minimum node size [42].

As mentioned above, Table 4 shows the hyperparameter space of RFs in detail.

**Table 4.** The hyperparameter space of RFs.

| Hyperparameter | Hyperparameter Candidates |
|---|---|
| The number of trees | minimal value: 100 |
| | maximal value: 300 |
| | numbers of steps: 100 |
| The size of the random subset of features | the number of features |
| | the square root of the number of features |
| | log base 2 of the number of features |
| Splitting criterion | mean squared error criterion |
| | mean absolute error criterion |
| The size of the tree | minimal value: 2 |
| | maximal value: 17 (the size of maximal tree) |
| | numbers of steps: 1 |
| Minimum node size | minimal value: 2 |
| | maximal value: 33 (10% of the number of training samples) |
| | numbers of steps: 1 |

*2.5. Evaluation Methods and Metrics*

We used 10-fold cross-validation on the training set and $R^2$ as an evaluation method and metric to implement feature selection and hyperparameter tuning, and we evaluated the model performance with mean squared error (MSE), root mean squared error (RMSE), and $R^2$ on the test set.

### 2.5.1. Tenfold Cross-Validation

The k-fold cross-validation process randomly splits the dataset into k roughly equal-sized folds and combines $k - 1$ folds into one training set for training model, while one fold is used as the validation set to test the model. This procedure is repeated k times, each time using a different fold as a validation set. Then, the result of the k-fold cross-validation is the average of values computed in the k iterations [21,26,35]. Furthermore, the typical values of k are 5 or 10 [45,46]. In short, we set the k value as 10, i.e., we used 10-fold cross-validation while implementing feature selection and hyperparameter tuning.

### 2.5.2. Evaluation Metrics

In our experiment, we evaluated the model performance with MSE, RMSE, and $R^2$, which are defined as [21]:

$$MSE = \frac{1}{n} \sum_{i=1}^{n} (y_i - \hat{y}_i)^2 \tag{7}$$

$$RMSE = \sqrt{\frac{1}{n} \sum_{i=1}^{n} (y_i - \hat{y}_i)^2} = \sqrt{MSE} \tag{8}$$

$$R^2 = 1 - \frac{\sum_{i=1}^{n}(y_i - \hat{y}_i)^2}{\sum_{i=1}^{n}(y_i - \overline{y})^2} = 1 - \frac{MSE}{MST} \tag{9}$$

where $n$ is the number of samples, $y_i$ is the true value of the $i$th sample, $\hat{y}_i$ is the predicted value of the $i$th sample, $\overline{y}$ is the mean value of the target vector, and MST is mean total sum of squares [21].

Specifically, the *MSE* and *RMSE* will have a value of 0 if the regression model fits the data perfectly [47], and the higher the values of MSE and RMSE, the worse the model will be [21]. Furthermore, the $R^2$ will have a value of 1 if the regression model fits the data perfectly, and the positive values of $R^2$ range from 0 to 1 [47], i.e., the closer to value 1, the better the model [21]. However, the $R^2$ will have negative value if the regression model performs poorly. As mentioned above, the best value and worst values of MSE, RMSE, and $R^2$ are shown in Table 5 [47].

**Table 5.** The best values and worst values of MSE, RMSE, and $R^2$.

| Metrics | Best Value | Worst Value |
| --- | --- | --- |
| MSE | 0 | $+\infty$ |
| RMSE | 0 | $+\infty$ |
| $R^2$ | $+1$ | $-\infty$ |

To summarize, an $R^2$ value of 0.8 clearly indicates very good regression model performance, and $R^2$ values such as 0.756 or 0.535 indicate good results [47].

### 2.6. Permutation Feature Importance

The permutation feature importance breaks the relationship between a feature and the target; therefore, it can be used to calculate the feature importance from the decrease in the model score. When the values of a feature are randomly shuffled, the link between the feature and the target is broken [39]. Furthermore, the permutation feature importance can be computed several times with different random permutations of a feature [31]. The permutation feature importance is defined as [31]:

$$i_j = s - \frac{1}{K} \sum_{k=1}^{K} s_{k,j} \tag{10}$$

where $i_j$ is the permutation feature importance of feature $f_j$, $s$ is the score of the model $m$ which is trained on the dataset $D$, $K$ is the number of different random permutations, and $s_{k,j}$ is the score of the model $m$ on data $\widetilde{D}_{k,j}$ (data $\widetilde{D}_{k,j}$ is generated by the randomly shuffling feature $f_j$ of the dataset $D$) [31].

In summary, we used $R^2$ as a metric to evaluate the score of the model, and the number of times to permute a feature was set to 100. Finally, we implemented the permutation feature importance on the test set, and the random_state parameter of the permutation_importance was set to 0.

## 3. Results and Discussion

### 3.1. Results of Feature Selection

Table 6 shows that the feature subset found by SFS was a lot better than the feature subset found by SBS. It improved the performance and reduced the dimensionality of the original dataset (from 34 to 7); therefore, we chose it to be the final feature subset for the KNN algorithm. However, the feature subset found by SBS failed to improve the performance, so we inferred that some relevant features might be eliminated in early iterations, which means that once those features are excluded, they cannot be included later, even though those features might possibly increase the performance [48]. After all, SBS has no guarantee of finding the optimal feature subset [27].

**Table 6.** Results of feature selection using KNN.

| Feature Set | Model Performance ($R^2$) | Number of Features |
|---|---|---|
| Original feature set | 0.3838 | 34 |
| The best feature subset found by SFS | 0.5709 | 7 |
| The best feature subset found by SBS | 0.3683 | 13 |

As shown in Table 7, the feature subset found by SFS is much better than the feature subset found by SBS. It improved the performance and reduced the dimensionality of the original dataset (from 34 to 7). Thus, we chose it to be the final feature subset for CARTs.

**Table 7.** Results of feature selection using CARTs.

| Feature Set | Model Performance ($R^2$) | Number of Features |
|---|---|---|
| Original feature set | 0.5354 | 34 |
| The best feature subset found by SFS | 0.7173 | 7 |
| The best feature subset found by SBS | 0.6140 | 3 |

As seen in Table 8, the best feature subset found by RFECV successfully improved the performance and reduced the dimensionality of the original dataset (from 34 to 13); therefore, it was selected to be the final feature subset for the RF.

**Table 8.** Results of feature selection using RF.

| Feature Set | Model Performance ($R^2$) | Number of Features |
|---|---|---|
| Original feature set | 0.6713 | 34 |
| The best feature subset found by RFECV | 0.7201 | 13 |

Table 9 shows all feature subsets selected after feature selection using different learning algorithms. Although the relevant features of those feature subsets are different, there are some common relevant features, including "the mental suffering damages that plaintiff claims" and "the age of the victim", which were selected by all of them. In addition, "the total number of plaintiffs who are victim's parents in the same case", "the total number of plaintiffs who are victim's children in the same case", "the defendant's income", and "the sum of mental suffering damages that all plaintiffs claim in the same case" were selected by two of them. Therefore, we can infer that those relevant features are crucial to the assessment of mental suffering damages in fatal car accident cases in Taiwan.

**Table 9.** The best feature subsets based on different learning algorithms.

| Serial Number | KNN | CART | RF |
|---|---|---|---|
| 1 | The total number of plaintiffs who are the victim's spouse in the same case | The total number of plaintiffs who are the victim's parents in the same case | The total number of plaintiffs who are the victim's parents in the same case |
| 2 | The age of the victim | The total number of plaintiffs who are the victim's children in the same case | The total number of plaintiffs who are the victim's children in the same case |
| 3 | The number of relatives that the plaintiff needs to support | The age of the victim | The total number of plaintiffs in the same case |
| 4 | The number of motorcycles that the plaintiff possesses | The number of children that the defendant raises | The age of the victim |
| 5 | The defendant's income | The number of investments that the defendant holds | The plaintiff is the victim's parent |
| 6 | The defendant belongs to disadvantaged and vulnerable groups | The sum of mental suffering damages that all plaintiffs claim in the same case | The plaintiff's educational background |
| 7 | The mental suffering damages that plaintiff claims | The mental suffering damages that plaintiff claims | The plaintiff's income |
| 8 | | | The amount of real estate that the plaintiff possesses |
| 9 | | | The defendant's educational background |
| 10 | | | The defendant's income |
| 11 | | | The amount of real estate that the defendant possesses |
| 12 | | | The sum of mental suffering damages that all plaintiffs claim in the same case |
| 13 | | | The mental suffering damages that the plaintiff claims |

*3.2. Results of Hyperparameter Tuning*

Table 10 shows that the generalization performance of KNN and RF improved after hyperparameter tuning, but the generalization performance of CART remains unchanged, which means that the combination of hyperparameters before tuning was already the best hyperparameter combination of CART. In addition, the best hyperparameter combinations of KNN, CART, and RF are listed in Table 11.

**Table 10.** Results of hyperparameter tuning using KNN, CART, and RF.

| Model | Performance | MSE | RMSE | $R^2$ |
|---|---|---|---|---|
| KNN | Before tuning | 1138.5934 | 33.7430 | 0.5709 |
| | After tuning | 762.1359 | 27.6068 | 0.7128 |
| CART | Before tuning | 750.0399 | 27.3868 | 0.7173 |
| | After tuning | 750.0399 | 27.3868 | 0.7173 |
| RF | Before tuning | 742.6653 | 27.2518 | 0.7201 |
| | After tuning | 728.9268 | 26.9986 | 0.7253 |

**Table 11.** The best hyperparameter combinations of KNN, CART, and RF.

| Model | Hyperparameter | Value of Hyperparameter |
|---|---|---|
| KNN | Distance metric | Manhattan distance |
| | The value of k | 5 |
| | Distance weighting | weighting by the inverse of neighbors' distance |
| CART | Splitting criterion | mean absolute error criterion |
| | The size of tree | 17 |
| | Minimum node size | 2 |
| RF | The number of trees | 200 |
| | The size of the random subset of features | the square root of the number of features |
| | Splitting criterion | mean squared error criterion |
| | The size of the tree | 16 |
| | Minimum node size | 2 |

*3.3. Final Performance Evaluation*

As seen in Table 12, the optimal KNN model performed very well on the training data, but the generalization performance of the optimal KNN model was worse than the performance on the training set. Therefore, it obviously suffered from overfitting. However, it cannot be denied that the optimal KNN model achieved a good $R^2$ score of 0.7128.

**Table 12.** The performance of the optimal KNN model.

| Model Performance | MSE | RMSE | $R^2$ |
|---|---|---|---|
| Training error | 77.8977 | 8.8259 | 0.9785 |
| Testing error (generalization performance) | 762.1359 | 27.6068 | 0.7128 |

As shown in Table 13, the optimal CART model performed well on the training set, whereas the generalization performance of the optimal CART model was not as good as the performance on the training data. Thus, it apparently suffered from overfitting. However, it cannot be denied that the optimal CART model achieved a good $R^2$ score of 0.7173.

**Table 13.** The performance of the optimal CART model.

| Model Performance | MSE | RMSE | $R^2$ |
|---|---|---|---|
| Training error | 87.2880 | 9.3428 | 0.9759 |
| Testing error (generalization performance) | 750.0399 | 27.3868 | 0.7173 |

Table 14 shows that the best RF model performed well on the training set; however, the generalization performance of the best RF model was not as good as the performance on the training set. Therefore, it suffered from overfitting. However, it cannot be denied that the best RF model achieved a good $R^2$ score of 0.7253.

**Table 14.** The performance of the optimal RF model.

| Model Performance | MSE | RMSE | $R^2$ |
|---|---|---|---|
| Training error | 122.6508 | 11.0747 | 0.9661 |
| Testing error (generalization performance) | 728.9268 | 26.9986 | 0.7253 |

Overall, Table 14 shows that the optimal RF model performed best and with the slightest overfitting among those optimal models, and the rankings of MSE, RMSE, and $R^2$ in Table 15 identically show that the optimal RF model was in the first position and was the top performing learning algorithm, as we expected at first. After all, ensemble learning is often considered to be more robust than a single base learner in practice [16]. Therefore, we choose it to be the final predictive model for plaintiffs, defendants, and lawyers, serving as a robust tool to predict the potential outcomes of judgments regarding discretionary damages of mental suffering in fatal car accident cases.

**Table 15.** The generalization performance and ranking of optimal KNN, CART, and RF models.

| Model | MSE | RMSE | $R^2$ |
|---|---|---|---|
| KNN | 762.1359 | 27.6068 | 0.7128 |
| CART | 750.0399 | 27.3868 | 0.7173 |
| RF | 728.9268 | 26.9986 | 0.7253 |
| Rankings | | | |
| 1st | RF | RF | RF |
| 2nd | CART | CART | CART |
| 3rd | KNN | KNN | KNN |

On the other hand, the optimal CART model and KNN model were ranked in second position and last position, respectively, although their performances were only slightly worse than that of the optimal RF model; therefore, these two models clearly show some potential in predicting discretionary damages for mental suffering in fatal car accident cases.

*3.4. Feature Importance Evaluation*

As seen in Figures 2–4 and Table 16, the most important feature is "the mental suffering damages that plaintiff claims", as we expected at first, and it corresponds with the results of feature selection, i.e., "the mental suffering damages that plaintiff claims" was selected by all models. In fact, this feature represents the disposition principle that all judges in Taiwan must abide by while assessing mental suffering damages, i.e., the assessment of mental suffering damages must be limited to the mental suffering damages that plaintiff claims. In a word, the final discretionary damages of mental suffering can only be equal to or lower than the mental suffering damages that the plaintiff claims. Therefore, the ML algorithms and permutation feature importance seem to be very clever and useful, because they automatically learn from the dataset and successfully reveal the important disposition principle in Taiwan without being taught by humans.

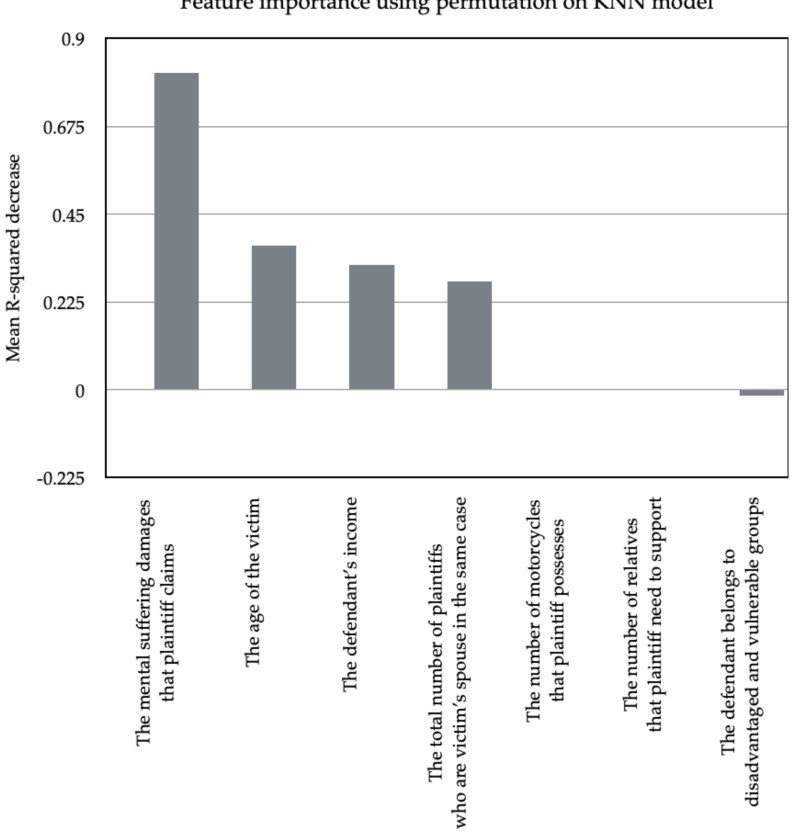

**Figure 2.** Feature importance using permutation on the KNN model.

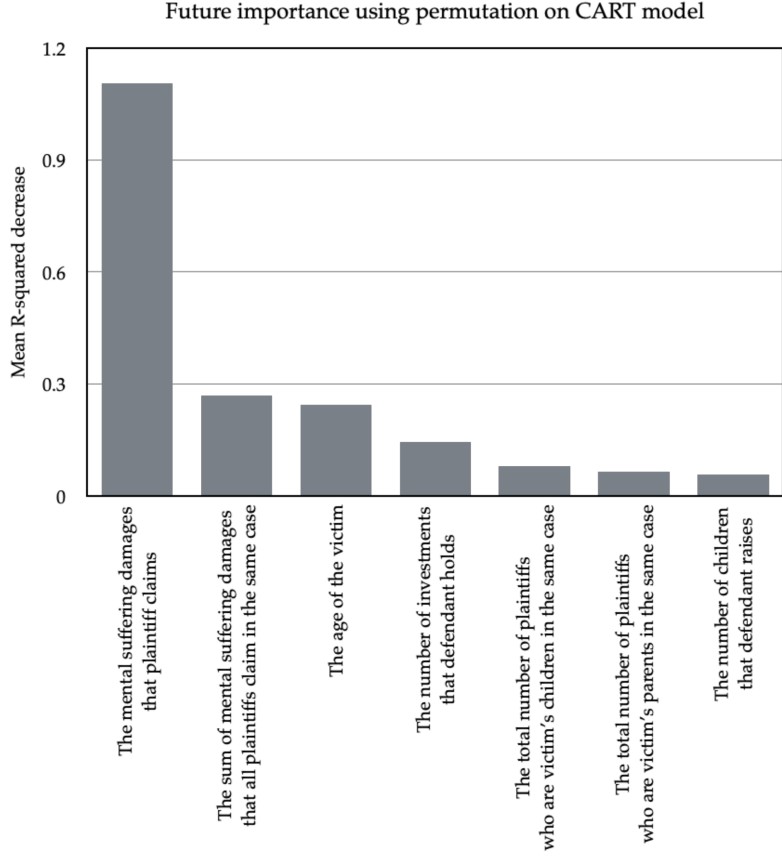

**Figure 3.** Feature importance using permutation on the CART model.

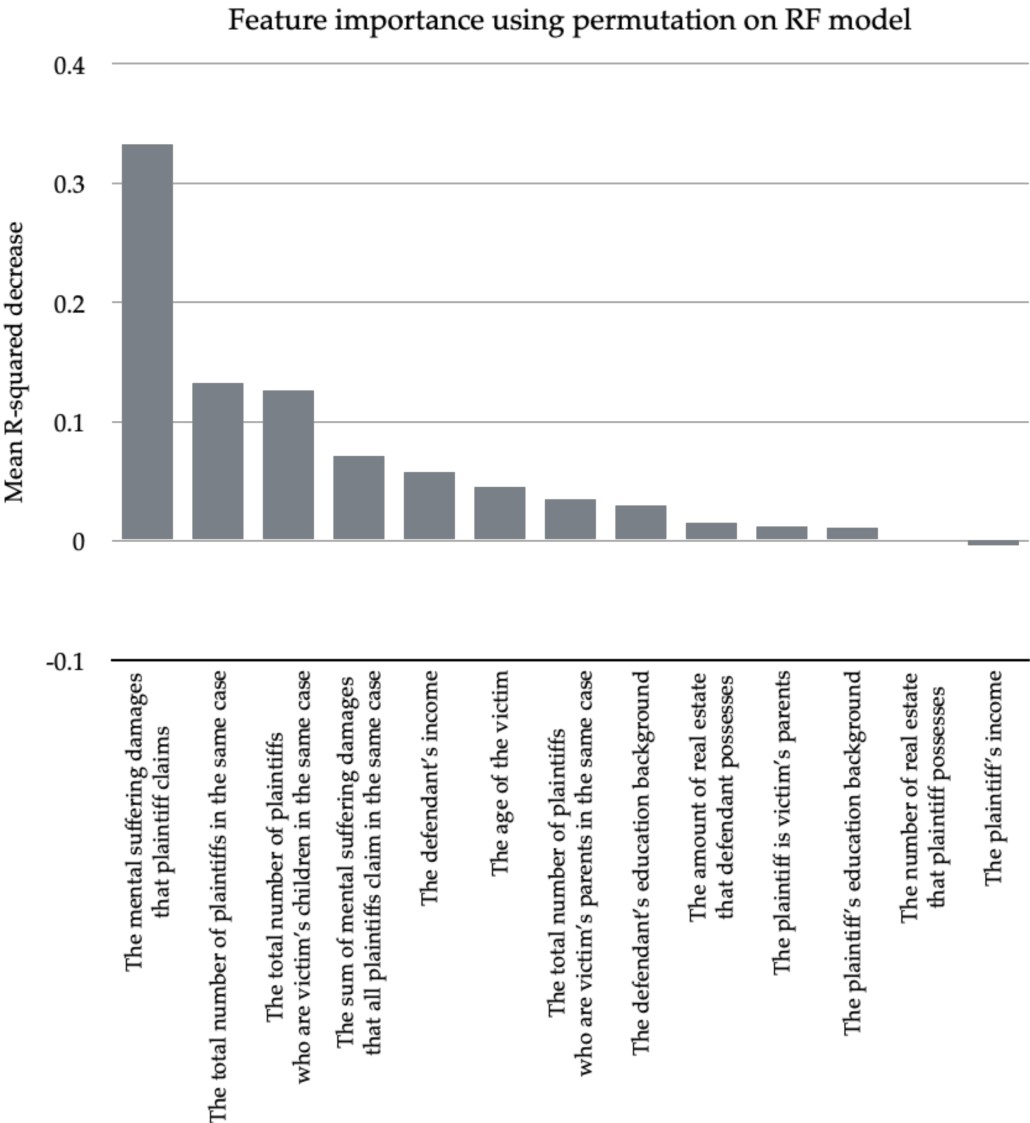

**Figure 4.** Feature importance using permutation on the RF model.

**Table 16.** Comparison of important features.

| Feature | KNN | CART | RF |
|---|---|---|---|
| The mental suffering damages that the plaintiff claims | 0.8105 | 1.1063 | 0.3327 |
| The age of the victim | 0.3685 | 0.2432 | 0.0447 |
| The defendant's income | 0.3205 | - | 0.0577 |
| The sum of mental suffering damages that all plaintiffs claim in the same case | - | 0.2680 | 0.0712 |
| The total number of plaintiffs who are the victim's parents in the same case | - | 0.0647 | 0.0343 |
| The total number of plaintiffs who are the victim's children in the same case | - | 0.0793 | 0.1254 |

Next, we focus on "the age of the victim", because it was selected by all models, and the ranking of that feature was high in the KNN and CART models. In practice, some judges indicate that the age of the victim will affect the level of mental suffering, and some studies have ascertained that the older the victim is, the lower the discretionary damages

of mental suffering will be [2]. Therefore, we may infer that the judges will consider the level of mental suffering to be slighter if the victim is older. In general, the death of a young person is often considered to be a pity according to public opinion, but the death of an older person, especially at a great age, is often considered to be a "happy death" or "enough", according to public opinion.

Moreover, "the defendant's income" was selected and found to be important by the KNN model, and it was also selected by the RF model; thus, we can infer that this feature will have a major effect on the discretionary damages of mental suffering. However, whether the defendant's income or earning capacity should be considered while assessing the mental suffering damages gives rise to much controversy. Some studies have established that the defendant's income or earning capacity is completely irrelevant to mental suffering damages [49], whereas some other studies have discerned that the defendant's income or earning capacity is completely relevant to the mental suffering damages and should be considered, pointing out that the higher the defendant's income or earning capacity, the higher the discretionary damages of mental suffering should be [17]. It is clear that judges in Taiwan may take the view and opinion of the latter, according to the ranking of feature importance we revealed.

Finally, "the total number of plaintiffs who are victim's parents in the same case", "the total number of plaintiffs who are victim's children in the same case", and "the sum of mental suffering damages that all plaintiffs claim in the same case" were selected by the CART and RF models; they were found to be important by the RF model, but only slightly important by the CART model. The reason why we built these features was to find out whether judges will reduce the discretionary damages of mental suffering in order to ease the burden on the defendant when there are multiple plaintiffs in the same case. Therefore, we may infer that our hypothesis is true.

In summary, "the mental suffering damages that plaintiff claims" and "the age of the victim" are key legal factors while assessing the mental suffering damages in fatal car accident cases in Taiwan. Apart from those features, "the defendant's income", "the total number of plaintiffs who are victim's parents in the same case", "the total number of plaintiffs who are victim's children in the same case", and "the sum of mental suffering damages that all plaintiffs claim in the same case" clearly indicate that judges are mainly concerned with whether the defendant is able to pay the mental suffering damages while assessing the mental suffering damages in fatal car accident cases in Taiwan.

## 4. Conclusions

To begin with, we successfully built an optimal regression model based on RF and achieved good performance, which can serve as a robust and professional tool for litigants and their lawyers, to predict the discretionary damages of mental suffering in advance; they can easily break the uncertainty of judicial outcome and wisely decide whether they should litigate or not. In addition, we revealed the importance ranking of legal factors, so legal actors can focus on the crucial factors such as "the mental suffering damages that plaintiff claims" and "the age of the victim" in order to develop the best litigation strategy. For example, they can concentrate on finding the evidentiary documents and materials concerning the most important or dominant legal factors instead of wasting precious time and money on irrelevant legal factors. Moreover, we also found that judges will reduce the discretionary damages of mental suffering in order to ease the burden on the defendant when there are multiple plaintiffs in the same case, and judges are mainly concerned with whether the defendant is able to pay the costs while assessing the mental suffering damages in fatal car accident cases in Taiwan. However, it cannot be denied that a good predictive model might possibly replace human judges, because it can easily make precise predictions about discretionary damages of mental suffering and maintain the coherence of discretionary damages of mental suffering. Similarly, the importance of lawyers might be lost because people can easily predict the potential outcomes of judgments and develop the best litigation strategy by themselves with the help of predictive models and the ranking

of legal factors, i.e., they can access high-quality and professional legal consulting services without expensive payment.

On the other hand, this methodology could be applied to other types of mental suffering damages prediction tasks in Taiwan with minor and suitable adjustments, including mental suffering from physical injuries or pain, humiliation, privacy infringements, and divorce. Moreover, people can claim mental suffering damages and judges need to consider several factors such as circumstances of illegal infringement and the earning capability of two parties in accordance with the Interpretation of the Supreme People's Court on Problems regarding the Ascertainment of Compensation Liability for Emotional Damages in Civil Torts [17], which is similar to the assessment of mental suffering damages in Taiwan; thus, our methodology could be applied to the region of mainland China after suitable adjustments, such as replacing the existing features with suitable legal factors.

Despite the achievements, there is still some work that can be done to make this research more significant. First of all, the dataset we used was too simple; therefore, we need to increase the amount of data. In addition, that dataset was made by legal experts and was full of quantitative features; it would be worthwhile extracting the features from the original judgment documents by NLP techniques to build a new dataset and examine whether the results can be improved or not. Although the optimal regression model based on RF achieved good performance with an $R^2 = 0.7253$, it is still far from "a very good performance with $R^2 = 0.8$". Therefore, these basic and classic ML algorithms (KNN, CART, and RF) apparently failed to overcome the obstacle to "a very good regression model with $R^2 = 0.8$"; thus, it is essential to experiment with advanced ML and DL algorithms such as SVM, gradient boosting, ANN, CNN, or an adaptive network-based fuzzy inference system (ANFIS) so that we might develop a very good regression model.

Above all, this study has successfully applied ML techniques to the prediction of discretionary damages of mental suffering in fatal car accidents in Taiwan and offers a very useful and handy tool to people and actors in the legal domain, as well as successfully solving long-standing problems and achieving good results. In addition, it also reveals the thinking and preferences ongoing in judges' minds.

**Author Contributions:** Conceptualization, D.H.; methodology, D.H.; software, D.H.; validation, D.H., L.C. and T.S.; formal analysis, D.H.; investigation, D.H.; resources, D.H.; data curation, D.H.; writing—original draft preparation, D.H.; writing—review and editing, D.H., L.C. and T.S.; visualization, D.H. and L.C.; supervision, D.H., L.C. and T.S.; project administration, D.H., L.C. and T.S. All authors have read and agreed to the published version of the manuscript.

**Funding:** This research received no external funding.

**Institutional Review Board Statement:** Not applicable.

**Informed Consent Statement:** Not applicable.

**Data Availability Statement:** Although the data we collected belong to legal issues, all judgments we used are open and available from the Taiwan Judicial Yuan Law and Regulations Retrieving System. The dataset we constructed and analyzed in this study can be found here: https://zenodo.org/record/5565766#.YW0X19lBwnV (accessed on 3 November 2021).

**Acknowledgments:** This research was partially supported by MOST-110-2221-E-260-015.

**Conflicts of Interest:** The authors declare no conflict of interest.

## Abbreviations

| | |
|---|---|
| ANN | artificial neural network |
| ANFIS | adaptive network-based fuzzy inference system |
| BERT | bidirectional encoder representations from transformer |
| CNN | convolutional neural network |
| CRF | conditional random field |
| DL | deep learning |
| GRU | gated recurrent unit |
| KNN | k-nearest neighbor |
| LSTM | long short-term memory |
| ML | machine learning |
| MSE | mean squared error |
| MST | mean total sum of squares |
| NLP | natural language processing |
| $R^2$ | R-squared |
| RFECV | recursive feature elimination with cross-validation |
| RMSE | root mean squared error |
| SBS | sequential backward selection |
| SFS | sequential forward selection |
| SVM | support vector machine |

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
