# Peer review of "Legal Judgment Prediction Based on Machine Learning: Predicting the Discretionary Damages of Mental Suffering in Fatal Car Accident Cases"

_applsci, doi:10.3390/app112110361_

Round 1

Reviewer 1 Report

The manuscript proposes RF, KNN, CART, and k-nearest neighbor for Legal Judgment Prediction. The comparative analysis had not been presented in a comprehensive manner. The other major issue is the lack of methods descriptions. No information had been shared on the methods mathematical and backgruond.  Although the legal judgment prediction using machine learning techniques has been a very important issue of artificial intelligence in legal domain the state of the art had not been comprehensive. The methods pesented in this work are too basic. What about hybrid methods, deep learning and other advanced ensembles? what is the reason behind proposing basic ML methods? Why the ANN and its variations, ANFIS, SVM are not discussed? Authors objective was to predict the discretionary damages of mental suffering in fatal car accident cases in Taiwan.Yet the data is not described in a sufficient manner. The ethical issue must be resolved and well described. The validation and comprative analysis of modeling with k-nearest neighbor, classification and regression trees, and random forests must be further elaborated.   Authors show that random forests optimal model outperforms the other models.Yet the question sis that how it can be further improved. What is the future direction.   It is intersting that lawyers can focus on the relevant and important legal factors without wasting time on the irrelevant legal factors with the help of the ranking of feature importance. However the ethical issue must be elaborated and discussed more.

Although the paper has appropriate length and informative content, several parts must be improved and written in better grammar and syntax. It would be essential if authors would consider revising the organization and composition of the manuscript, in terms of the definition/justification of the objectives, description of the method, the accomplishment of the objective, and results. The paper is generally difficult to follow. Paragraphs and sentences are not well connected. Furthermore, I advise considering using standard keywords to better present the research. Remove the keywords to ML/DL methods, instead use the standard keywords not more than 5. Please revise the abstract according to the journal guideline. It must be under 200 words.

The research question, method, and the results must be briefly communicated. The abstract must be more informative. I suggest having four paragraphs in the introduction for; describing the concept, research gap, contribution, and the organization of the paper. The motivation has the potential to be more elaborated. You may add materials on why doing this research is essential, and what this article would add to the current knowledge, etc.

The originality of the paper is not discussed well. The research question must be clearly given in the introduction, in addition to some words on the testable hypothesis. Please elaborate on the importance of this work. Please discuss if the paper suitable for broad international interest and applications or better suited for the local application? Elaborate and discuss this in the introduction.

State of the art needs improvement. A detailed description of the cited references is essential. Several recently published papers are not included in the review section. In fact, the acknowledgment of the past related work by others, in the reference list, is not sufficient. Consequently, the contribution of the paper is not clear. Furthermore, consider elaborating on the suitability of the paper and relevance to the journal. Kindly note that references cited must be up to date.    

Elaborate on the method used and why used this method.

Limitations and validation are not discussed adequately. The research question and hypothesis must be answered and discussed clearly in the discussion and conclusions. Please communicate the future research. The lessons learned must be further elaborated in the conclusion by discussing the results to the community and the future impacts. What is your perspective on future research?     Insert acronyms and abbreviation table. Equations are not cited properly Several claims are not cited properly.

Author Response

Dear Sir/ Madam,

        Thank you for spending much of your precious time reviewing my manuscript, please see the attachment.

Reviewer 2 Report

In this paper, the authors: (a) develop a dataset for predicting discretionary damages of mental suffering, and (b) train and compare the predictive power of multiple machine learning algorithms, using an existing ML library for Python, scikit-learn. The paper is very much application focused, and overall fits well with the theme of the journal Applied Sciences. The ML algorithms used in the paper are not new. However, the application is a novel use of ML. Overall, the paper is written well, although some minor improvements can be made to the language. I have mostly minor issues among my detailed feedback below. The first one is critical as it is a missing citation for the article that introduced the ML library that the authors used in their experimental study.

1) At the beginning of Section 2, you are missing a citation for scikit-learn. Please see this link, https://scikit-learn.org/stable/about.html#citing-scikit-learn, where the authors of scikit-learn provide the citation information for the paper about the library. Specifically, you should cite this paper:

Scikit-learn: Machine Learning in Python, Pedregosa et al., JMLR 12, pp. 2825-2830, 2011.

The BibTex entry for it is:

@article{scikit-learn,
 title={Scikit-learn: Machine Learning in {P}ython},
 author={Pedregosa, F. and Varoquaux, G. and Gramfort, A. and Michel, V.
         and Thirion, B. and Grisel, O. and Blondel, M. and Prettenhofer, P.
         and Weiss, R. and Dubourg, V. and Vanderplas, J. and Passos, A. and
         Cournapeau, D. and Brucher, M. and Perrot, M. and Duchesnay, E.},
 journal={Journal of Machine Learning Research},
 volume={12},
 pages={2825--2830},
 year={2011}
}

2) Although the citation above is a must, I also recommend including a URL to the website for the package, https://scikit-learn.org/, perhaps as a footnote.

3) You have created a very interesting dataset. You should ensure that the very dataset that you create and used in your experiments remains available in the future. Although you provide a link to the dataset via Google Drive, I have concerns over its long-term availability via that mechanism. I strongly recommend archiving the dataset with a service such as Zenodo, and/or the Harvard Dataverse, or something similar. Many of these services even issue a DOI for the dataset, including for specific versions in the event that the dataset is revised in the future. The ones I mentioned here are even free. You might also consider submitting a "data paper" such as to MDPI's Data. But personally, I think your manuscript, as it stands is already a good source for the description of the dataset, and I would find it more valuable to just archive the data with Zenodo, the Harvard Dataverse, or something similar.

4) Many of your section headings are currently in all lowercase, primarily the headings of subsubsections. Please check these, and correct casing.

5) Equations 3 and 4 share some notation, but you end up defining that notation twice, once on line 150 and then again on line 153. Since these 2 equations are so close within the paper, I'd recommend finding a way to just define the shared notation a single time to avoid the redundancy.

6) The writing quality overall is good. There are a few errors here and there, so make sure you proofread the paper. The small number of such errors are all minor, such as a spelling error in the section 3.1 heading.

Author Response

Dear Sir/ Madam,

Thank you for spending much of your precious time reviewing my manuscript, Please see the attachment.

Reviewer 3 Report

This paper presents machine learning-based models for legal judgment prediction, particularly for predicting discretionary damages of mental suffering in fatal car accident cases in Taiwan. To predict accurate legal judgment, the authors perform feature and model optimization. The comments for the paper are as follows.

  1. The paper has insignificant contributions. Simply speaking, the paper’s contribution in the current form is only to predict discretionary mental damages with popular machine learning algorithms (somewhat outdated in the era of deep learning) and a simple dataset. Considering recent studies such as legal judgment prediction with deep learning or text-based legal judgment prediction, the novelty of this paper seems insufficient. The authors should find a way to enhancing and emphasizing their contributions (by adding more descriptions, experimental results, discussions, and anything). For example, what is different from existing models?, Why should those algorithms be used compared to other algorithms such as boosting and deep learning models?, How much does the proposed model improve the accuracy?, What new insight did the authors discover?, Is the proposed method applicable to other cases except for Taiwan?, etc.

  1. The texts of the paper are segmented into too many paragraphs, which reduces the readability of the paper rather than improve it.

  1. The abstract of the paper is too lengthy. In particular, lines 22-28 seem unsuitable for the abstract. The authors should revise this part.

  1. In equation (4), the equation, median(y)_m=median(y) y∈Q_m, seems unnecessary. Furthermore, why are m and Q_m used to indicate the same thing? The authors should define Q_m (e.g. a set of data samples in node m) or use only one of them.

  1. The sentence in lines 191-192 is unclear. Does “all nominal categorical features” include features described in the previous paragraph?

  1. The settings of hyper-parameter have something wrong. First, random_state is not the model hyper-parameter, so finding the optimal value of random_state is unnecessary. Second, in most cases, the size of the bootstrap sample generally is unchanged hyper-parameter. Third, RF is also a tree-based model so the hyper-parameters of trees used for RF should be considered like CART(The size of the tree, Splitting criterion, and so on).

  1. According to tables 5 and 6, SBS degrades the model performance rather than improves. Why does such a problem happen? The authors should state the reason.

  1. SBS and SFS can be used for RF. But, the authors use RFECV instead of them. Is RFECV better than both methods? The authors should present the reason why RFECV should be used for RF by the related references or experimental results.

  1. In table 11, MAE increases after hyper-parameter tuning, but the authors do not present the reason.

  1. In lines 411-412, is R-squared score of 0.7128 a good result? What is criterion on that?

  1. Although the authors present the results of measuring feature importance, they do not discuss them in detail.

Author Response

(The authors gave the same response as above.)

Round 2

Reviewer 1 Report

The comments had been all addressed.

Author Response

Dear Sir/ Madam,

Thank you for spending much of your precious time reviewing our manuscript. 

Best regards,
DeCheng Hsieh

Reviewer 3 Report

The authors dealt with my concerns well.
Thus, I agree to accept this paper in the present form.

But, text editing in some revised parts seems to be required.
For instance, some abbreviations in the introduction were used without definition.

Author Response

Dear Sir/ Madam,

Thank you for spending much of your precious time reviewing our manuscript. We have revised the mistakes according to your advice.

Best regards,
DeCheng Hsieh